# Difficulties and Psychological Impact of the SARS-CoV-2 Pandemic in Patients with Systemic Lupus Erythematosus: A Nationwide Patient Association Study

**DOI:** 10.3390/healthcare10020330

**Published:** 2022-02-09

**Authors:** Marc Scherlinger, Naimah Zein, Jacques-Eric Gottenberg, Marianne Rivière, Jean-François Kleinmann, Jean Sibilia, Laurent Arnaud

**Affiliations:** 1Rheumatology Department, Centre Hospitalier Universitaire de Strasbourg, 1 Avenue Molière, 67098 Strasbourg, France; jacques-Eric.gottenberg@chru-strasbourg.fr (J.-E.G.); jean-francois.kleinmann@chru-strasbourg.fr (J.-F.K.); jean.sibilia@chru-strasbourg.fr (J.S.); laurent.arnaud@chru-strasbourg.fr (L.A.); 2Centre National de Référence des Maladies Auto-Immunes et Systémiques Rares Est/Sud-Ouest (RESO), 67000 Strasbourg, France; 3Division of Rheumatology and Clinical Immunology, Beth Israel Deaconess Medical Center, Harvard Medical School, Boston, MA 02115, USA; 4Association Française du Lupus et Autres Maladies Auto-Immunes (AFL+), 4 Rue Lafayette, 57000 Metz, France; naimah.zein@hotmail.fr (N.Z.); lupusplus@gmail.com (M.R.); 5Immunologie, Immunopathologie et Chimie Thérapeutique (I2CT)-UPR3572 CNRS, Institut de Biologie Moléculaire et Cellulaire (IBMC), 67084 Strasbourg, France; 6Laboratoire d’Immuno Rhumatologie Moléculaire, Institut National de la Santé et de la Recherche Médicale (INSERM) UMR_S 1109, Institut Thématique Interdisciplinaire (ITI) de Médecine de Précision de Strasbourg, Transplantex NG, Faculté de Médecine, Fédération Hospitalo-Universitaire OMICARE, Fédération de Médecine Translationnelle de Strasbourg (FMTS), Université de Strasbourg, 67084 Strasbourg, France

**Keywords:** systemic lupus erythematosus, SARS-CoV-2, pandemic, patient perspective

## Abstract

Objectives: We aimed to evaluate the difficulties encountered by systemic lupus erythematosus (SLE) patients during the early COVID-19 pandemic and to evaluate their impact on patient mental health. Methods: We conducted a nationwide survey including SLE patients from France, recruited by their treating specialist or through a patient association. The survey was administered online or in paper form between November 2020 and April 2021 and included questions aiming at evaluating the difficulties encountered during the early COVID-19 pandemic (March to August 2020). The impact on mental health was evaluated using the Hospital Anxiety and Depression Scale (HADS) and the Post-Traumatic Stress Disorder (PTSD) Checklist for DSM-5 (PCL-5). Results: 536 SLE patients (91.9% women) of mean age 50 (±14.1) years responded to the survey. The main reported difficulties were issues regarding access to medical care (*n* = 136, 25.4%) or hydroxychloroquine treatment (*n* = 98/389, 25.2%), the loss of employment (*n* = 85/349, 24.4%), and financial difficulties (*n* = 75/536, 11%). In 328 patients with complete mental health assessments, 161 (47.2%) screened positive for anxiety, 141 (41.2%) screened positive for depressive syndrome, and 128 (38.7%) screened positive for PTSD. The multivariate analysis showed that female sex (OR = 4.29 [95%CI: 1.39–13.24]), financial issues (OR = 2.57 [1.27–5.22]), and difficulties accessing medical care (OR = 2.15 [1.26–3.69]) or hydroxychloroquine treatment (OR = 1.90 [1.06–3.40]) were independently associated with a positive screening for PTSD. Conclusions: The COVID-19 pandemic resulted in a severe burden in SLE patients, including difficulties accessing care and treatment along with high psychological distress. Better understanding these difficulties will allow for better prevention and care in times of crisis.

## 1. Introduction

The emergence of SARS-CoV-2 in late 2019 led to a worldwide pandemic affecting hundreds of millions of people and is responsible for an excess mortality of at least 5 million people worldwide [1]. Initially, few data were available concerning the risk factors for COVID-19-related mortality. Glucocorticoids and immunosuppressive drugs were considered major risk factors of severe COVID-19 [2]. Therefore, most patients with Systemic Lupus Erythematosus (SLE) were considered at risk and needed to be extremely careful during the SARS-CoV-2 pandemic. These patients were recommended to self-limit their normal and social activities to decrease their risk of contracting COVID-19, a strategy also known as shielding. Furthermore, an early (but unfounded) craze for hydroxychloroquine (HCQ) as a potential COVID-19 treatment led to a rush toward its limited stock, making SLE patients at risk of running out of medications [3] and susceptible to disease flare-up [4].

The conjunction of these factors, together with consecutive lockdowns and uncertainties linked to the COVID-19 pandemic, were major hardships for SLE patients. Considering the baseline negative impact of SLE on patients’ quality of life and psychological health [5], the COVID-19 pandemic may have taken a huge toll on the psychological health of these patients. Indeed, a Danish study conducted on more than 5000 patients with autoimmune diseases identified difficulties with shielding at work as major determinants of anxiety in these patients [6]. An evaluation of the psychological impact of a global pandemic on SLE patients is important for better tailoring patient care in times of hardships.

The objective of this study was to evaluate the cumulative difficulties and psychological burden of the COVID-19 pandemic in SLE patients at a nation-wide level in France.

## 2. Methods

### 2.1. Patients and Ethical Consideration

Patients with a self-reported diagnosis of SLE were recruited in Metropolitan France and French overseas departments and territories with the help of the patients association AFL plus and their treating specialist. All patients received an information notice and accepted the anonymous use of their data. The current study named “EPICURE” received approval from Strasbourg’s University Hospital ethical committee (CE-2020-151).

### 2.2. Survey Design

The survey was conducted both online (SurveyLegend©, Malmö, Sweden) and using paper questionnaires (with prepaid return envelopes) and was distributed between November 2020 and April 2021. The patients were recruited by their treating specialist or directly through the patient association “Association France Lupus et autres maladies auto-immunes, AFL+”. The survey was designed for French-speaking patients and aimed at evaluating demographic and lupus-specific characteristics. Difficulties encountered during the early COVID-19 pandemic (March to August 2020) were evaluated, including issues with obtaining treatment or seeking medical advice. Screening for anxiety, depression, and post-traumatic stress disorder were conducted using a validated translation [7,8] of the Hospital Anxiety and Depression Scale (HADS-A/D) and Post-Traumatic Stress Disorder (PTSD) Checklist 5 (PCL-5). Screening was considered positive for anxiety when HADS-Anxiety ≥ 10, for depression when HADS-Depression ≥ 7 [7], and for post-traumatic stress disorder when PCL-5 ≥ 31 [8].

### 2.3. Statistical Analysis

The quantitative data are reported as the mean with standard deviation and were compared using Student’s t-test. The qualitative variables are reported as percentages and compared using the chi-square test. The multivariate model (logistic regression) was constructed using a backward stepwise selection of variables that were associated with the endpoint in univariate analysis. The statistical analysis was conducted using STATA 13.0 (Statacorp, College Station, TX, USA). All tests were two-sided, and a *p*-value < 0.05 was considered statistically significant.

## 3. Results

### 3.1. Socio-Demographic and SLE Characteristics of Respondents

We gathered 536 questionnaires (online, *n* = 334; paper questionnaire, *n* = 202) from individual SLE patients between November 2020 and April 2021. The mean age was 50 (±14.1), years and 91.9% were women. SLE had been diagnosed 17.3 (±11) years earlier, 88.1% (472/536) had cutaneo-articular manifestations, and 22.6% (121/536) had kidney involvement. Of the patients, 77.2% (389/504) were treated with HCQ, 50.1% (247/493) were treated with corticosteroids, and 33.3% (167/496) were treated with immunosuppressive drugs. Other patient characteristics are shown in Table 1. Except for a younger age, the characteristics of online responders were similar to those of paper responders.

### 3.2. Impact of the COVID-19 Pandemic

Following the first wave of COVID-19, 275 patients (79% of the non-retired/student/disabled population, *n* = 349) carried on with their professional activities either at their workplace (*n* = 125) or by teleworking (*n* = 150). A total of 85 (24.3%) patients discontinued their professional activity (technical unemployment (*n* = 55) and lost jobs (*n* = 30)). A preventive sick leave was prescribed to 92 patients. Financial difficulties were reported by 76 patients (14.9%) and were significantly more prevalent in those who had discontinued their professional activity (36.9% vs. 9.3%, *p* < 0.0001). The patients reported that their most reliable source of health information during the pandemic (scored ≥ 4 on a 5 points scale) were patient associations (51%) followed by their lupus specialist (32.2%), their general practitioner (28.7%), and finally mass media (28.2%).

### 3.3. Healthcare during the COVID-19 Pandemic

Corticosteroids and immunosuppressive drugs were tapered in 16.2% (40/247) and 12% (20/167) of cases, respectively (Table 2). Tapering was conducted at the initiative of the lupus specialist in 65% of cases (immunosuppressive drug: 58%) or by the patient in 22.5% of cases (immunosuppressive drug: 37.5%). Hydroxychloroquine shortage was responsible for difficulties in obtaining the drug for 25.2% (98/389) of HCQ-treated patients, and 57 had to interrupt HCQ treatment for a median of 7 days (IQR: 3–17).

During the early pandemic, 157 patients (31.5%) reported a SLE flare. Flare were more prevalent among patients who tapered corticosteroid compared with those who did not (55% vs. 26.1%, *p* < 0.001). Conversely, interruption of HCQ due to shortage did not affect flare rate (32.1% vs. 30.6%, *p* = 0.82). Difficulties accessing medical care (general practitioner or specialist) was reported by 136 patients (25.4%), and 25.3% (127/503) of patients were able to teleconsult with their lupus specialist. Difficulties consulting their lupus specialists were reported by 19% (97/510) patient and were significantly lower in those who were able to teleconsult (33% vs. 67%, *p* < 0.05).

### 3.4. Psychological Evaluation

Sleeping issues were reported by 54.2% (272/502) of patients. Psychological tests were all completed for 328 patients (61.2%; Figure 1). Screening for anxiety and depression were positive for 47.6% and 41.5%, respectively. Additionally, 39% of patient screened positive for PTSD. Among 204 patients with at least one positive screening, 46 (22.6%) sought the help of a psychiatrist and 23 were prescribed anxiety- or depression-related medication. Factors associated with mental health condition in univariate analysis are shown in Appendix A. Multivariate analysis showed that female sex (OR = 4.29 [95%CI: 1.39–13.24]), financial issues (OR = 2.57 [1.27–5.22]), and difficulties accessing medical care (OR = 2.15 [1.26–3.69]) or hydroxychloroquine treatment (OR = 1.90 [1.06–3.40]) were independently associated with a positive screening for PTSD (Table 3). Concerning anxiety, difficulties accessing medical care was the only significantly associated factor (OR = 1.94 (1.15 3.25), *p* = 0.012). Financial difficulties and difficulties accessing medical care were significantly associated with a positive screening for depression (Table 3). Prevalence of suspected anxiety, depression, and PTSD as well as reported difficulties were similar between metropolitan and overseas departments of France as well as between highly and mildly impacted areas.

## 4. Discussion

This survey including 536 patients with SLE in France demonstrates the difficulties encountered during the early SARS-CoV-2 pandemic. We found that 85 patients discontinued their professional activity, resulting in financial issues in more than a third of them. Another major difficulty identified was the HCQ shortage, which affected almost a quarter of HCQ-treated patients, sometime leading to its temporary discontinuation. In line with these findings, a high prevalence of psychiatric disorders was reported, including anxiety (47.6%), depression (41.5%), and post-traumatic stress disorder (39%). Importantly, we found that female sex, financial issues, and difficulty obtaining HCQ or medical advice were independently associated with PTSD.

A meta-analysis reported that the pooled prevalence of anxiety was 31.9% (95%CI: 27.5–36.7%), and that of depression was 33.7% (27.5–40.6%) in the general population during the pandemic [9]. Although this remains an indirect comparison, we found a higher prevalence of anxiety and depression in SLE patients compared with the general population. In an European study of SLE patients conducted before the pandemic, the prevalence of anxiety was 30.5% and that of depression was 15.3% [10]. Other studies conducted during the pandemic were heterogeneous: one from Turkey showed that prevalence of depression (43%) and PTSD (28%) but not anxiety (20%) was similar to our results [11], while another from Argentinia showed very low prevalence of anxiety (9.6%) or depression (13.2%) [12]. These discrepancies might be explained by the use of different tools to measure psychological burden as well as geographical differences. Increased psychiatric burden was partly expected because, during the early pandemic, patients with SLE were believed to be at high risk of severe COVID-19 and faced many challenges (e.g., issues obtaining HCQ or medical advice) [13]. For example, a study conducted by the patient association Lupus Europe found that HCQ shortage was a major driver of SLE patient anxiety and that this effect persisted long after the initial shortage [14]. Interestingly, recent studies have shed light on the importance of stress/mental health on the development of autoimmune diseases as well as on the risk of flare [15,16]. However, a striking result was that less a quarter of patients who screened positive for a mental health disorder sought professional help. This finding may be explained by the absence of a diagnosis (lack of screening or self-conscience of such a diagnosis) or by the difficulty in seeking medical help due to the pandemic. Collectively, these results underline the importance of screening mental health disorders to offer proper management.

Patient associations provide educational materials and help them cope with their disease. In our study, the majority of the respondents reported patient associations as the main source of health information during the pandemic. Furthermore, patient associations bear an important role for advocating patient rights. As an example, the French patient association AFL-plus recognized the emergency that arose from limited HCQ access for patients with rheumatic disease, necessitating measures taken to prevent shortage. Coordinated communication with the authorities led to a nationwide decree reserving HCQ to patients with underlying autoimmune diseases. As a result, the median duration of HCQ discontinuation due to shortage was only 7 days, limiting its impact on the patients disease activity since HCQ half-life is 40 days. It expected that, without this legal response, the impact of HCQ shortage on SLE patient health might have been catastrophic. In case of prolonged HCQ shortage, strategies aiming at temporarily halving HCQ doses could potentially limit the impact of shortage [17].

The main limitations of our study are related to its design as a survey, which exposes a study to participation bias and may limit the representativity of the population included. To prevent this bias, we recruited patients from lupus specialists and from patient’s association. Moreover, this design allowed us to recruit at a national level including overseas departments. To limit self-reporting/memorization bias, patients were asked about recent events (early pandemic) and the mental health assessment relied on validated self-administrated questionnaires. Finally, the design did not allow us to accurately estimate the participation rate of the study.

## 5. Conclusions

In conclusion, the COVID-19 pandemic was associated with several difficulties significantly impacting the mental health of patients with SLE. Better recognition of these factors are necessary to better prevent, screen for, and treat mental disorders during times of crisis.

## Figures and Tables

**Figure 1 healthcare-10-00330-f001:**
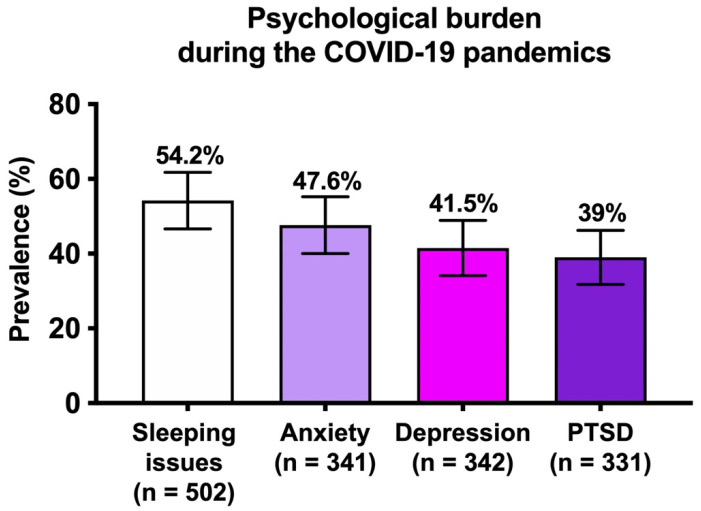
Prevalence of psychiatric disorders in SLE patients during the COVID-19 pandemic. Columns show mean and bars represent the standard deviation. Abbreviation: PTSD, post-traumatic stress disorder.

**Table 1 healthcare-10-00330-t001:** Socio-demographical characteristics of the survey respondents.

	SLE Patients (*n* = 536)
Age, mean (±S.D.)	50 (±14.1)
Female sex, *n* (%)	489 (91.9%)
Employment status, *n* (%)	
- Student	15 (2.8%)
- Employed	291 (54.3%)
- Unemployed	107 (20%)
- Disability	50 (46.7%)
- Retired	109 (20.3%)
- Prefer not to answer	14 (2.6%)
Study level, *n* (%)	
- Pre-high school	51 (9.5%)
- High school	202 (37.7%)
- Bachelor	209 (39%)
- Master or doctorate	74 (13.8%)
Marital status, *n* (%)	
- Single	103 (19.2%)
- Married/concubine	370 (69%)
- Divorced	56 (10.5%)
- Prefer not to answer	7 (1.3%)
Time since lupus diagnosis, mean years (±S.D.)	17.3 (±11)
Lupus organ involvement, *n* (%)	
- Cutaneo-articular	472 (88.1%)
- Lupus nephritis	121 (22.6%)
Lupus treatments, *n/N* (%)	
- Hydroxychloroquine	389/504 (77.2%)
- Glucocorticoids	247/493 (50.1%)
- Immunosuppressant	167/496 (33.3%)

**Table 2 healthcare-10-00330-t002:** Difficulties reported during the early COVID-19 pandemic. Abbreviation: GP, general practitioner; SLE, systemic lupus erythematosus.

	SLE patients (*n* = 536)
Corticosteroid tapering, *n/N* (%)	40/247 (16.2%)
Immunosuppressive drug tapering, *n/N* (%)	20/167 (12%)
Reported SLE flare-up, *n/N* (%)	157/498 (31.5%)
Difficulty to obtain hydroxychloroquine, *n/N* (%)	98/389 (25.2%)
- Interruption of HCQ due to shortage, *n/N* (%)	57/389 (14.7%)
- Duration of interruption, days, median (IQR)	7 (3–17)
Difficulties accessing medical care, *n/N* (%)	136/510 (25.4%)
- Difficulties consulting GP, *n/N* (%)	77/519 (14.8%)
- Difficulties consulting lupus specialist, *n/N* (%)	97/510 (19%)
Teleconsultation	
- Teleconsultation with GP, *n/N* (%)	157/513 (30.6%)
- Teleconsultation with lupus specialist, *n/N* (%)	127/503 (25.3%)

**Table 3 healthcare-10-00330-t003:** Factors associated with the development of anxiety, depression, or PTSD. Odds ratio (CI95%) using multivariate logistic regression are shown.

Patient and Pandemic-Associated Factors	Odds Ratio (95%CI) for Anxiety	Odds Ratio (95%CI) for Depression	Odds Ratio (95%CI) for PTSD
Female sex	2.25 (0.97–5.25) *p* = 0.058	ns	4.29 (1.39–13.24) *p* = 0.01
Financial difficulties	ns	2.59 (1.31–5.11) *p* = 0.006	2.57 (1.27–5.22) *p* = 0.009
Difficulties obtaining HCQ	1.70 (0.97–2.98) *p* = 0.065	ns	1.90 (1.06–3.40) *p* = 0.03
Difficulties accessing medical care	1.94 (1.15 3.25) *p* = 0.012	2.57 (1.53–4.33) *p* < 0.0001	2.15 (1.26–3.69) *p* = 0.005

## Data Availability

Data may be made available by reasonable request to the lead contact.

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
