# Peer review of "Difficulties and Psychological Impact of the SARS-CoV-2 Pandemic in Patients with Systemic Lupus Erythematosus: A Nationwide Patient Association Study"

_healthcare, 2022, doi:10.3390/healthcare10020330_

Round 1
Reviewer 1 Report
The article submitted by Scherlinger et al. aims to study the psychological consequences of the COVID19 pandemic in patients with systemic lupus erythematosus in France through an online survey. The authors showed that 47.6% of the patients developed signs of anxiety, 41.5% of depression and 39% of post traumatic stress disorder with some factors associated with their development.
General Comment:
The manuscript is clear, well-structured, and sheds light on the issue of the psychological consequences of the COVID19 pandemic in patients with chronic illness.
One of the main limitations is related to the experimental design since the data are collected via an open-ended online survey exposing to a participation bias. Indeed, patients had a self reported diagnosis of SLE and were recruted by patients association which may not reflect the general population of SLE patients. In particular, a large proportion of patients in this study were treated with corticosteroids, which is not usual in SLE patients, and health information was reported to come mainly from patients associations which was not expected. This point could be adressed int he discussion part.
Apart from that, the results are interesting and show the impact and potential solutions to limit the psychological burden of the COVID19 pandemic.
Specific comments
Methods (line 72): What was the proportion of paper questionnaires given by the specialist compared to the online questionnaire and does this impact the results?
Line 91: What was the response rate (not sure it is possible if the study was available online) ?
Results:
General comment:
Have you seen differences between regions, especially for the region most affected by the pandemic during the first wave in France
Specific comments
Line 108: Did the source of the information impact the psychological burden (mass media could generate more stress) ?
Table 2: others parameters used in the univariate analysis could be interesting to show here. Especially, Lupus flare, COVID-19 disease or teleconsultation were they associated with psychological disorder ?
Discussion
line 152: The authors may mention the percentage of anxiety or depression seen in the SLE patient population outside the pandemic to highlight the impact of the pandemic.
Author Response
Reviewer 1 :
The article submitted by Scherlinger et al. aims to study the psychological consequences of the COVID19 pandemic in patients with systemic lupus erythematosus in France through an online survey. The authors showed that 47.6% of the patients developed signs of anxiety, 41.5% of depression and 39% of post traumatic stress disorder with some factors associated with their development.
General Comment:
The manuscript is clear, well-structured, and sheds light on the issue of the psychological consequences of the COVID19 pandemic in patients with chronic illness.
One of the main limitations is related to the experimental design since the data are collected via an open-ended online survey exposing to a participation bias. Indeed, patients had a self reported diagnosis of SLE and were recruted by patients association which may not reflect the general population of SLE patients. In particular, a large proportion of patients in this study were treated with corticosteroids, which is not usual in SLE patients, and health information was reported to come mainly from patients associations which was not expected. This point could be adressed int he discussion part.
Apart from that, the results are interesting and show the impact and potential solutions to limit the psychological burden of the COVID19 pandemic.
We thank the Reviewer for his positive assessment of our study. We agree with the Reviewer and we have modified the discussion section to discuss the Reviewer’s points (see below). We also agree that generally-speaking, too many patients are treated with glucocorticoids (50.1% in our study). When comparing with similar studies such as an European survey on 4375 SLE patients, 52.3% of them were treated with glucocorticoids (doi:10.1136/ lupus-2020-000469) which is similar to our study.
Page 7, lines 205-212: “The main limitations of our study are related to its design as a survey which exposes to a participation bias and may limit the representativity of the including population. To prevent this bias we recruited patients from lupus specialists and from patients association. Moreover this design allowed to recruit at a national level including overseas departments. To limit the self-reporting/memorization bias, patients were asked about recent events (early pandemic) and the mental health assessment relied on validated self-administrated questionnaires. Finally, the design did not allow us to accurately estimate the participation rate of the study.”
Specific comments
Methods (line 72): What was the proportion of paper questionnaires given by the specialist compared to the online questionnaire and does this impact the results?
We thank the Reviewer for this question. There was 334 online responses (62.3%) and 202 paper responses (27.7%). As expected, online respondents were younger than paper ones (mean 47.3 +/- 13.3 vs. 54.5 +/- 14.3, p < 0.001). Other patient characteristics including difficulties during the pandemic and most importantly, psychological tests results were similar between these 2 groups. As suggested, we have added a comment on this in the results section.
Page 3, line 97: “We gathered 536 questionnaires (online, n = 334; paper questionnaire, n = 202) from individual SLE patients between November 2020 and April 2021.”
Page 3, line 103-104: “Except for a younger age, online responders characteristics were similar to paper responders. ”
Line 91: What was the response rate (not sure it is possible if the study was available online) ?
As the Reviewer suggests, our study design does not allow us to evaluate study response rate or the proportion of questionnaire that were given by the specialist. Indeed the paper questionnaire were distributed by clinician but also among patients by patient association and sent back anonymously. We have added this point in the limitation section.
Page 7, line 211-212: ”Finally, the design did not allow us to accurately estimate the participation rate of the study.”
Results:
General comment:
Have you seen differences between regions, especially for the region most affected by the pandemic during the first wave in France
The Reviewer raises an excellent point that we did not evaluate. To address this comment, we compared the prevalence of anxiety, depression and PTSD between regions most affected by the first wave in France (Grand-Est, Haut-de-France, and Ile de France, n = 233) and the others (n = 296). We did not find statistical difference. We have added this point in the results section.
Page 5 line s149-152: “Prevalence of suspected anxiety, depression and PTSD as well as reported difficulties were similar between metropolitan and overseas department of France, as well as between highly and mildly impacted areas.”
Specific comments
Line 108: Did the source of the information impact the psychological burden (mass media could generate more stress) ?
To address this point we compared the psychological burden between patients who used mass media as a source of reliable information (score of >= 4 on a 5 scale). We did not find any statistical difference in term of depression, anxiety or PTSD.
Table 2: others parameters used in the univariate analysis could be interesting to show here. Especially, Lupus flare, COVID-19 disease or teleconsultation were they associated with psychological disorder ?
We thank the reviewer for this suggestion. To address the Reviewer’s point, we have added a supplementary table 1 to the submitted revised manuscript. This table includes the univariate analysis of several variables including those requested by the Reviewer. As the Reviewer can see, flare is significantly associated (p < 0.05) with depression and PTSD in univariate analysis. However, significance was lost in multivariate analysis, therefore we did not add flare in the multivariate model.
Supplementary table 1: Factors associated with the development of anxiety, depression or PTSD in univariate analysis (logistic regression).
|
|
Odds ratio (95%CI) for anxiety |
Odds ratio (95%CI) for depression |
Odds ratio (95%CI) for PTSD |
|
Difficulties to obtain HCQ |
2.02 (1.17-3.50) p = 0.01 |
1.62 (0.95-2.77) p = 0.07 |
2.26 (1.30-3.92) p = 0.004 |
|
Difficulties to access medical care |
1.97 (1.20-3.25) p = 0.008 |
2.64 (1.60-4.38) p < 0.001 |
2.16 (1.30-3.60) p = 0.003 |
|
Financial difficulties |
1.41 (0.74-2.69) p = 0.30 |
2.62 (1.35-5.10), p = 0.005 |
2.57 (1.30-5.08) p = 0.007 |
|
Female Sex |
1.99 (0.91-4.37) p = 0.085 |
1.31 (0.60-2.82) p = 0.49 |
3.30 (1.22-8.88) P = 0.018 |
|
Lupus flare |
1.59 (1.00-2.52) p = 0.05 |
1.64 (1.03-2.60) p = 0.04 |
1.66 (1.03-2.66) p = 0.04 |
|
Reported COVID-19 disease |
0.96 (0.50-1.82) p = 0.89 |
0.73 (0.37-1.42) p = 0.35 |
1.03 (0.53-1.98) p = 0.93 |
|
Use of teleconsultation |
1.04 (0.68-1.61) p = 0.85 |
0.94 (0.6-1.45) p = 0.77 |
0.82 (0.52-1.28) p = 0.38 |
Discussion
line 152: The authors may mention the percentage of anxiety or depression seen in the SLE patient population outside the pandemic to highlight the impact of the pandemic.
In accordance with the Reviewer suggestion, we have added a reference evaluating the prevalence of anxiety and depression in SLE patients before the pandemics.
Page 6 lines 174-175: “In an European study of SLE patients conducted before the pandemic, the prevalence of anxiety was 30.5% and depression 15.3%[10].“
Reviewer 2 Report
Dear Authors! Thank you for the opportunity to review the manuscript. The aim of the study is clear and useful for practicing physicians. The manuscript is well-written and good-organized. Minor English revisions are required.
Author Response
Reviewer 2 :
Dear Authors! Thank you for the opportunity to review the manuscript. The aim of the study is clear and useful for practicing physicians. The manuscript is well-written and good-organized. Minor English revisions are required.
We thank the Reviewer for his positive assessment of our study.
Reviewer 3 Report
The authors report data from a nationwide survey from France and French territories in SLE patients concerning psychological effects. Their main findings reported are that four factors have a major impact on PTSD: female sex, financial issues, HCQ access, and access to medical care. By contrast, depression was associated with financial issues and access to medical care.
Aside from the inherent limitations of a survey (which should be stated more clearly), that is, self-reporting bias, I have a couple of points that need to be addressed:
A. The introduction is quite short but covers the main aspects. However, more emphasis could be placed on the possible psychological effects.
B. Methods section. Here, several points need attention.
- The distribution of the survey is not entirely clear: how many patients were approached (how? paper-based? via email? other means?). I doubt that it would be possible in a survey to obtain informed consent. Maybe there was some passage in the survey?
- survey design: it would be interesting to state the survey questions in detail in a supplement. the authors state that difficulties encountered early in the pandemic were evaluated (March 2020 to August 2020).
- The authors used translated versions of questionnaires. Evaluated translations? Translations by the authors? more details required.
- more details on STATA required (location of the corp.).
C. Results
- is it possible to report the proportion of returned questionnaires based on the distribution of the survey? or was this done online and cant be tracked back.
- of possible interest (and it would also add some novelty) is whether there were differences between metropolitan France and overseas territories as access to medical care / medications may differ
- one major point: authors report results between Nov 2020 and april 2021. This contrasts with the methods section (March to August 2020). please clarify
- Table 1: More details on lupus organ involvement would be interesting other than LN vs cutaneo-articular. Same for treatments. Also, please homogenize reporting: n (%) for all categories
- Figure 1: n of patients should also be stated here, also please report positive and negative SD
- Results reported in section 3.3 would be suitable for an additional table
- Major point: the results reported in Section 3.4 diverge from those in table 2! please clarify (example: text female sex OR 3.7 (1.03-13.49) for PTSD; table 4.28 (1.39-13.24)!
- also pertaining to the results from the multivariate analysis: if the authors used backward stepwise regression, results from the univariate analysis should also be stated (could be combined in table 2).
D. Discussion
- the discussion is quite short and I think that there are several important references missing.
- First, limitations (and also the advantages) of the survey design should be discussed more in-depth
- some references to consider for discussion as a couple of surveys have been published on the topic (PMID): 32586918, 32616604, 32669305, 32927376, 34459303
- also, to discuss the impact of halting HCQ on flares: A Pharmacokinetics-Informed Approach to Navigating Hydroxychloroquine Shortages in Patients With Rheumatic Disease During the COVID-19 Pandemic. ACR Open Rheumatol. 2020 Aug;2(8):491-495. doi: 10.1002/acr2.11164. Epub 2020 Jul 29.PMID: 32725866
Author Response
Reviewer 3 :
The authors report data from a nationwide survey from France and French territories in SLE patients concerning psychological effects. Their main findings reported are that four factors have a major impact on PTSD: female sex, financial issues, HCQ access, and access to medical care. By contrast, depression was associated with financial issues and access to medical care.
Aside from the inherent limitations of a survey (which should be stated more clearly), that is, self-reporting bias, I have a couple of points that need to be addressed:
We thank the Reviewer for his assessment of our work. We have added a sentence to discuss self-reporting bias in the limitation section of the discussion.
Page 7, lines 205-212: “The main limitations of our study are related to its design as a survey which exposes to a participation bias and may limit the representativity of the including population. To prevent this bias we recruited patients from lupus specialists and from patients association. Moreover this design allowed to recruit at a national level including overseas departments. To limit the self-reporting/memorization bias, patients were asked about recent events (early pandemic) and the mental health assessment relied on validated self-administrated questionnaires. Finally, the design did not allow us to accurately estimate the participation rate of the study.”
- The introduction is quite short but covers the main aspects. However, more emphasis could be placed on the possible psychological effects.
We thank the Reviewer for his positive assessment of our introduction. We have amended the introduction to place more emphasis on the possible psychological effects.
Page 2 lines 56-62: “Considering the baseline negative impact of SLE on patients quality of life and psychological health [5], the COVID-19 pandemic may have taken a huge toll on the psychological health of these patients. Indeed, a Danish study conducted on more than 5000 patients with autoimmune diseases identified difficulties to shield at work as major determinants of anxiety in these patients [6]. The evaluation of the psychological impact of a global pandemic on SLE patients is important for better tailoring patient care in times of hardships.“
- Methods section. Here, several points need attention.
- The distribution of the survey is not entirely clear: how many patients were approached (how? paper-based? via email? other means?). I doubt that it would be possible in a survey to obtain informed consent. Maybe there was some passage in the survey?
We thank the Reviewer for raising this point. Patients were approached in a multimodal manner: from their treating specialist during a normal follow-up consult, from patient association through their usual mean to communicate to patients (their website, patients meetings) and by directly mailing the questionnaire to their members together with a prepaid return envelope. This multimodal approach hinders the possibility to accurately estimate the number of approached patients (and therefore the response rate).
In accordance with the Reviewer’s suggestion, we have modified the methods section to better describe these points.
Page 2 lines 74-78: “The survey was conducted both online (SurveyLegend© platform) and using paper questionnaires (with prepaid return envelope) and was distributed between November 2020 and April 2021. The patients were recruited by their treating specialist or directly through the patient association “Association France Lupus et autres maladies auto-immunes, AFL+”.”
Additionally, the patients all received an information notice and gave a non-opposition to use their (anonymous) data by submitting their questionnaire. We have added the information notice as well as the questionnaire as a supplementary file. We have also changed the methods section to better describe this.
Page 2 lines 70-71: “All patients received an information notice and accepted the anonymous use of their data.”
- survey design: it would be interesting to state the survey questions in detail in a supplement. the authors state that difficulties encountered early in the pandemic were evaluated (March 2020 to August 2020).
In accordance with the Reviewer’s suggestion, we have added the questionnaire and the patient information notice as a supplementary file.
- The authors used translated versions of questionnaires. Evaluated translations? Translations by the authors? more details required.
The Reviewer is right, we have used validated translation of these questionnaires. Please find below references of validation.
- HADS/A/D (reference 7): Roberge P. A psychometric evaluation of the French-Canadian version of the Hospital Anxiety and Depression Scale in a large primary care population. J Affect Disord 2013;:9.
- PCL-5 (reference 8): Ashbaugh AR, Houle-Johnson S, Herbert C, et al. Psychometric Validation of the English and French Versions of the Posttraumatic Stress Disorder Checklist for DSM-5 (PCL-5). PLoS One. 2016 Oct 10;11(10):e0161645. doi: 10.1371/journal.pone.0161645.
We have modified the methods to state this more clearly, page 2 line 82: “Screening for anxiety, depression and post-traumatic stress disorder were conducted using a validated translation (7,8) of the Hospital Anxiety and Depression Scale (HADS-A/D) and Post-traumatic stress disorder (PTSD) Checklist 5 (PCL-5).“
- more details on STATA required (location of the corp.).
We have added the required details on STATA.
Page 2, line 93: ”Statistical analysis were conducted using STATA 13.0 (Statacorp, College Station, Texas, USA).”
- Results
- is it possible to report the proportion of returned questionnaires based on the distribution of the survey? or was this done online and cant be tracked back.
We thank the Reviewer for raising this point that was also raised by the other Reviewers. Patients were approached in a multimodal manner: from their treating specialist during a normal follow-up consult, from patient association through their usual mean to communicate to patients (their website, patients meetings) and by directly mailing the questionnaire to their members together with a prepaid return envelope. This multimodal approach hinders the possibility to accurately estimate the number of approached patients (and therefore the response rate).
We have added this limitation in the discussion section page 7, lines 211-212: “Finally, the design did not allow us to accurately estimate the participation rate of the study.”
- of possible interest (and it would also add some novelty) is whether there were differences between metropolitan France and overseas territories as access to medical care / medications may differ.
We thank the Reviewer for suggesting this comparison that we did not think of.
In patients living in oversea territories (n = 41) compared to patients living in metropolitan France, we did not find statistical differences in the prevalence of anxiety (oversea: 32.1% vs metropolitan: 48.5% p =0.12), depression (oversea: 42.8% vs metropolitan: 41.1 p = 0.84), PTSD (oversea: 34.6% vs metropolitan: 39% p = 0.83), difficulty of access to hydroxychloroquine (oversea: 25.6% vs metropolitan: 17.7% p =0.22) or difficulty to access to medical care (oversea: 36.6% vs metropolitan: 25.8% p = 0.14).
We have added this point in the results section (page 5, lines 149-152): “Prevalence of suspected anxiety, depression and PTSD as well as reported difficulties were similar between metropolitan and oversea departments of France, as well as between highly and mildly impacted areas.“
- one major point: authors report results between Nov 2020 and april 2021. This contrasts with the methods section (March to August 2020). please clarify
We thank the Reviewer for raising this point. The survey was conducted between November 2020 and April 2021. However, the questions were aimed at evaluating the difficulties lived during the first lockdown/ severe restrictions in France, thus from March to early August of 2020.
We have modified the abstract and methods section to make this more easier for the reader:
Abstract :” The survey was administered online or in paper form between November 2020 and April 2021, and included questions aiming at evaluating the difficulties encountered during the early COVID-19 pandemic (March to August 2020).”
Methods page 2 lines 73-81: “The survey was conducted both online (SurveyLegend© platform) and using paper questionnaires (with prepaid return envelope) and was distributed between November 2020 and April 2021. (…) Difficulties encountered during the early COVID-19 pandemic (March to August 2020) were evaluated, including issues to obtain treatment or seek medical advice. “
- Table 1: More details on lupus organ involvement would be interesting other than LN vs cutaneo-articular. Same for treatments. Also, please homogenize reporting: n (%) for all categories
We agree with the Reviewer that more information about organ involvement would be interesting. Since this was a very long questionnaire, we initially chose to focus the questions on difficulties and psychological burden of the pandemic.
In accordance with the Reviewer’s suggestion we have homogenized the Table 1 by reporting n (%).
Table 1: Socio-demographical characteristics of the survey respondents.
|
|
SLE patients (N = 536) |
|
Age, mean (±S.D.) |
50 (±14.1) |
|
Female sex, n (%) |
489 (91.9%) |
|
Employment status, n (%) - Student - Employed - Unemployed o Disability - Retired - Prefer not to answer |
15 (2.8%) 291 (54.3%) 107 (20%) 50 (46.7%) 109 (20.3%) 14 (2.6%) |
|
Study level, n (%) - Pre-highschool - Highschool - Bachelor - Master or doctorate |
51 (9.5%) 202 (37.7%) 209 (39%) 74 (13.8%) |
|
Marital status, n (%) - Single - Married/concubine - Divorced - Prefer not to answer |
103 (19.2%) 370 (69%) 56 (10.5%) 7 (1.3%) |
|
Time since lupus diagnosis, mean years (±S.D.) |
17.3 (±11) |
|
Lupus organ involvement, n (%) - Cutaneo-articular - Lupus nephritis |
472 (88.1%) 121 (22.6%) |
|
Lupus treatments, n/N (%) - Hydroxychloroquine - Glucocorticoids - Immunosuppressant |
389/504 (77.2%) 247/493 (50.1%) 167/496 (33.3%) |
- Figure 1: n of patients should also be stated here, also please report positive and negative SD
We thank the Reviewer for this suggestion. We have edited the figure in accordance with the Reviewer suggestion.
- Results reported in section 3.3 would be suitable for an additional table
We thank the Reviewer for this suggestion. We have added a Table (numbered 2, below) with the results from section 3.3 and renumbered initial Table 2 in Table 3 in the revised manuscript.
Table 2: Difficulties reported during the early COVID-19 pandemic. Abbreviation: GP, general practitioner; SLE, systemic lupus erythematosus.
|
|
SLE patients (N = 536) |
|
Corticosteroid tapering, n/N (%) |
40/247 (16.2%) |
|
Immunosuppressive drug tapering, n/N (%) |
20/167 (12%) |
|
Reported SLE flare-up, n/N (%) |
157/498 (31.5%) |
|
Difficulty to obtain hydroxychloroquine, n/N (%) - Interruption of HCQ due to shortage, n/N (%) - Duration of interruption, days, median (IQR) |
98/389 (25.2%) 57/389 (14.7%) 7 (3-17) |
|
Difficulties to access medical care, n/N (%) - Difficulties to consult GP, n/N (%) - Difficulties to consult lupus specialist, n/N (%) |
136/510 (25.4%) 77/519 (14.8%) 97/510 (19%) |
|
Teleconsultation - Teleconsultation with GP, n/N (%) - Teleconsultation with lupus specialist, n/N (%) |
157/513 (30.6%) 127/503 (25.3%) |
- Major point: the results reported in Section 3.4 diverge from those in table 2! please clarify (example: text female sex OR 3.7 (1.03-13.49) for PTSD; table 4.28 (1.39-13.24)!
We thank the Reviewer for raising this point that was also raised by another Reviewer. The results given in the abstract and Table 2 are the correct ones. We have modified the results section accordingly.
Page 5, lines 143-146: “Multivariate analysis showed that female sex (OR=4.29 [95%CI: 1.39-13.24]), financial issues (OR=2.57 [1.27-5.22]), difficulties to access medical care (OR=2.15 [1.26-3.69]), or to obtain hydroxychloroquine treatment (OR=1.90 [1.06-3.40]) were independently associated with a positive screening for PTSD (Table 3).”
- also pertaining to the results from the multivariate analysis: if the authors used backward stepwise regression, results from the univariate analysis should also be stated (could be combined in table 2).
We thank the Reviewer for this suggestion. To address the Reviewer’s point, we have added a supplementary table 1 with the univariate analysis of several variables including those requested by the Reviewer 1.
Supplementary table 1: Factors associated with the development of anxiety, depression or PTSD in univariate analysis (logistic regression).
|
|
Odds ratio (95%CI) for anxiety |
Odds ratio (95%CI) for depression |
Odds ratio (95%CI) for PTSD |
|
Difficulties to obtain HCQ |
2.02 (1.17-3.50) p = 0.01 |
1.62 (0.95-2.77) p = 0.07 |
2.26 (1.30-3.92) p = 0.004 |
|
Difficulties to access medical care |
1.97 (1.20-3.25) p = 0.008 |
2.64 (1.60-4.38) p < 0.001 |
2.16 (1.30-3.60) p = 0.003 |
|
Financial difficulties |
1.41 (0.74-2.69) p = 0.30 |
2.62 (1.35-5.10), p = 0.005 |
2.57 (1.30-5.08) p = 0.007 |
|
Female Sex |
1.99 (0.91-4.37) p = 0.085 |
1.31 (0.60-2.82) p = 0.49 |
3.30 (1.22-8.88) P = 0.018 |
|
Lupus flare |
1.59 (1.00-2.52) p = 0.05 |
1.64 (1.03-2.60) p = 0.04 |
1.66 (1.03-2.66) p = 0.04 |
|
Reported COVID-19 disease |
0.96 (0.50-1.82) p = 0.89 |
0.73 (0.37-1.42) p = 0.35 |
1.03 (0.53-1.98) p = 0.93 |
|
Use of teleconsultation |
1.04 (0.68-1.61) p = 0.85 |
0.94 (0.6-1.45) p = 0.77 |
0.82 (0.52-1.28) p = 0.38 |
- Discussion
- the discussion is quite short and I think that there are several important references missing.
- First, limitations (and also the advantages) of the survey design should be discussed more in-depth.
We thank the Reviewer for this comment that was also raised by other Reviewers. We have extended the paragraph on the limitations of the study and our strategies to limit these limitations.
Page 7, lines 205-212: “The main limitations of our study are related to its design as a survey which exposes to a participation bias and may limit the representativity of the including population. To prevent this bias we recruited patients from lupus specialists and from patients association. Moreover this design allowed to recruit at a national level including overseas departments. To limit the self-reporting/memorization bias, patients were asked about recent events (early pandemic) and the mental health assessment relied on validated self-administrated questionnaires. Finally, the design did not allow us to accurately estimate the participation rate of the study.”
- some references to consider for discussion as a couple of surveys have been published on the topic (PMID): 32586918, 32616604, 32669305, 32927376, 34459303
We thank the Reviewer for these suggestions. We have included several more surveys conducted recently in the discussion in order to better compare our results with the literature.
Page 6, lines 174-180: ” In an European study of SLE patients conducted before the pandemic, the prevalence of anxiety was 30.5% and depression 15.3%[10]. Other studies conducted during the pandemic were heterogeneous: one from Turkey showed that prevalence of depression (43%) and PTSD (28%) but not anxiety (20%) were similar to our results [11], while another from Argentinia showed very low prevalence of anxiety (9.6%) or depression (13.2%) [12]. These discrepancies might be explained by the use of different tools to measure psychological burden as well as geographical differences.”
- also, to discuss the impact of halting HCQ on flares: A Pharmacokinetics-Informed Approach to Navigating Hydroxychloroquine Shortages in Patients With Rheumatic Disease During the COVID-19 Pandemic. Scheetz MH, Konig MF, Robinson PC, Sparks JA, Kim AHJ.ACR Open Rheumatol. 2020 Aug;2(8):491-495. doi: 10.1002/acr2.11164. Epub 2020 Jul 29.PMID: 32725866
We thank the Reviewer for this suggestion that we have implemented in the discussion section.
Page 7, lines 203-204: “In case of prolonged HCQ shortage, strategies aiming at temporarily halving HCQ doses could potentially limit the impact of shortage [17].”
Reviewer 4 Report
- What is the response rate of this survey
- The period the survey was administered was not mentioned in method section although it was stated in abstract
- Why did you perform multivariate analysis to identify predictors for only PTSD? Why did you not perform multivariate analysis to identify predictors for depression, anxiety, and sleeping issues? I saw analysis for depression and anxiety in Table 2 but the result was not described in result section
- “Multivariate analysis showed that female sex (OR = 3.7, [1.03-13.48]), financial issues (OR = 2.98 [1.38-6.45]), or difficulties in obtaining HCQ (OR = 2.14 [1.01-4.52]) or medical care (OR = 2.15 [1.26-3.69] were independent predictors of positive PTSD screening test. The OR in this statement was not consistent with ones reported in Table 2 and abstract
- I would suggest not to use purple background, particularly the dark purple, in table. It is difficult to read the text. Just use all white background is more neat and professional
Author Response
Reviewer 4:
- What is the response rate of this survey
We thank the reviewer for his assessment of our work. Patients were approached in a multimodal manner: from their treating specialist during a normal follow-up consult, from patient association through their usual mean to communicate to patients (their website, patients meetings) and by directly mailing the questionnaire to their members together with a prepaid return envelope. This multimodal approach hinders the possibility to accurately estimate the number of approached patients (and therefore the response rate).
Unfortunately our study design does not allow us to evaluate study response. We have added this limitation in the discussion section.
Page 7, lines 211-212: “Finally, the design did not allow us to accurately estimate the participation rate of the study.”
- The period the survey was administered was not mentioned in method section although it was stated in abstract
We thank the Reviewer for pointing out this discrepancy. We have added the study period in the methods of the revised manuscript.
Page 2, lines 74-76: “The survey was conducted both online (SurveyLegend© platform) and using paper questionnaires (with prepaid return envelope) and was distributed between November 2020 and April 2021.“
- Why did you perform multivariate analysis to identify predictors for only PTSD? Why did you not perform multivariate analysis to identify predictors for depression, anxiety, and sleeping issues? I saw analysis for depression and anxiety in Table 2 but the result was not described in result section
We thank the Reviewer for his comment. In accordance with the Reviewer’s suggestion We have now described these findings in the text of the results section.
Page 5, lines 146-149: “Concerning anxiety, difficulties to access medical care was the only significantly associated factor (OR=1.94 (1.15 3.25), p = 0.012]). Financial difficulties and difficulties to access medical care were significantly associated with a positive screening for depression (Table 3).”
- “Multivariate analysis showed that female sex (OR = 3.7, [1.03-13.48]), financial issues (OR = 2.98 [1.38-6.45]), or difficulties in obtaining HCQ (OR = 2.14 [1.01-4.52]) or medical care (OR = 2.15 [1.26-3.69] were independent predictors of positive PTSD screening test. The OR in this statement was not consistent with ones reported in Table 2 and abstract.
We thank the Reviewer for pointing this out. We have checked all the calculations again and the results given in the abstract and Table 2 were the correct ones. We have modified the results section accordingly.
Page 5, lines 143-146: “Multivariate analysis showed that female sex (OR=4.29 [95%CI: 1.39-13.24]), financial issues (OR=2.57 [1.27-5.22]), difficulties to access medical care (OR=2.15 [1.26-3.69]), or to obtain hydroxychloroquine treatment (OR=1.90 [1.06-3.40]) were independently associated with a positive screening for PTSD (Table 3).”
- I would suggest not to use purple background, particularly the dark purple, in table. It is difficult to read the text. Just use all white background is more neat and professional
As suggested by the Reviewer, we have modified the Table style to use white and grey colors (please see below as an example). We are open that the Journal may ultimately change the Table style to fit with its own style.
Table 3 : Factors associated with the development of anxiety, depression or PTSD. Odds ratio (95%CI) using multivariate logistic regression are shown.
|
Patient and pandemic-associated factors |
Odds ratio (95%CI) for anxiety |
Odds ratio (95%CI) for depression |
Odds ratio (95%CI) for PTSD |
|
Female sex |
2.25 (0.97-5.25), p = 0.058 |
ns |
4.29 (1.39-13.24) p = 0.01 |
|
Financial difficulties |
ns |
2.59 (1.31-5.11) p = 0.006 |
2.57 (1.27-5.22) p = 0.009 |
|
Difficulties to obtain HCQ |
1.70 (0.97-2.98) p = 0.065 |
ns |
1.90 (1.06-3.40) p = 0.03 |
|
Difficulties to access medical care |
1.94 (1.15 3.25) p = 0.012 |
2.57 (1.53-4.33) p < 0.0001 |
2.15 (1.26-3.69) p = 0.005 |
Round 2
Reviewer 3 Report
The authors have addressed all major points. Only one suggestion regarding figure 1 remains: i suggest to add the n of patients to the respective bars to give the reader also the absolute numbers (not only of the total cohort ) in addition to the percentages.
Author Response
We thank the Reviewer for his positive assessment of our revised manuscript. We have modified figure 1 in accordance with the Reviewer’s suggestion (please see below).

Reviewer 4 Report
all of my comments have been addressed
Author Response
We thank the Reviewer for his positive assessment of our revised manuscript.
This manuscript is a resubmission of an earlier submission. The following is a list of the peer review reports and author responses from that submission.